# Capturing additional genetic risk from family history for improved polygenic risk prediction

Tianyuan Lu [1,2✉], Vincenzo Forgetta[1], J. Brent Richards [1,3,4,5] & Celia M. T. Greenwood [1,3,4,6✉]

Family history of complex traits may reflect transmitted rare pathogenic variants, intra-familial shared exposures to environmental and lifestyle factors, as well as a common genetic predisposition. We developed a latent factor model to quantify trait heritability in excess of that captured by a common variant-based polygenic risk score, but inferable from family history. For 941 children in the Avon Longitudinal Study of Parents and Children cohort, a joint predictor combining a polygenic risk score for height and mid-parental height was able to explain ~55% of the total variance in sex-adjusted adult height z-scores, close to the estimated heritability. Marginal yet consistent risk prediction improvements were also achieved among ~400,000 European ancestry participants for 11 complex diseases in the UK Biobank. Our work showcases a paradigm for risk calculation, and supports incorporation of family history into polygenic risk score-based genetic risk prediction models.

[1] Lady Davis Institute for Medical Research, Jewish General Hospital, Montreal, QC, Canada. [2] Quantitative Life Sciences Program, McGill University, Montreal, QC, Canada. [3] Department of Epidemiology, Biostatistics and Occupational Health, McGill University, Montreal, QC, Canada. [4] Department of Human Genetics, McGill University, Montreal, QC, Canada. [5] Department of Twin Research and Genetic Epidemiology, King's College London, London, UK. [6] Gerald Bronfman Department of Oncology, McGill University, Montreal, QC, Canada. ✉email: tianyuan.lu@mail.mcgill.ca; celia.greenwood@mcgill.ca

Predicting phenotypic values of complex traits, or the risks and outcomes of complex diseases have strong implications in health care and biomedical research[1,2]. In recent years, large-scale genome-wide association studies (GWASs) have characterized the genetic architecture of many complex traits and diseases[3]. Developing polygenic risk scores aggregating the effects of well-profiled genetic determinants has become possible[4,5]. Polygenic risk scores have demonstrated the potential to improve risk stratification in large populations[6–9], assist diagnosis and clinical differentiation[10–12], and refine risk management and treatment strategies[13–15].

However, most polygenic risk scores only capture linear additive effects of common genetic variants. Currently, most GWASs—even those based on the largest biobank studies—restrict consideration to genetic variants with a minor allele frequency > 0.1% or higher[3,16]. Furthermore, extremely rare pathogenic variants with high penetrance are rarely detected with array-based genotyping and imputation[3,17]. Furthermore, powerful approaches to accurately model more complex non-linear effects (i.e. dominance effects) and interaction effects (i.e. gene-by-environment effects and gene-by-gene effects) in a high dimensional setting are scarce, particularly since these effects are in general weaker than the linear additive effects[3,18]. Despite continuing methodological innovations in mining hidden heritability, these under-captured genetic effects likely prevent polygenic risk scores from achieving a further-improved predictive performance.

In contrast, family history information, such as parental measures of phenotypic values and disease records, provides an indirect measure of the overall genetic predisposition among relatives[19]. Although traditionally considered as a crucial risk factor for Mendelian diseases, family history has also shown added value in polygenic risk prediction[20–24]. For instance, individuals at an elevated level of risk for various types of cancer and cardiovascular diseases are more likely to be identified by assessments combining the family history of disease with polygenic risk scores[20–24], compared to using polygenic risk scores or family history alone.

Nevertheless, creating accurate joint predictors may be challenging, since it requires modelling individual-level training data on phenotypes, genotypes, and family history information, as has been explored previously[7,25–27]. This may not lead to effective prediction models if datasets containing all the required information are too small, particularly for diseases with a low prevalence in the population.

Therefore, in this work, we demonstrated both by theory and with examples, the improved predictions associated with a scheme for combining polygenic risk scores and parental disease histories. We approached this goal by inquiring what proportion of trait variance is captured by parental information and not by existing polygenic risk scores. Then, in comparison to using polygenic risk scores alone or predictors based only on parental trait measures or parental disease history, we evaluated the performance of these joint predictors in predicting adult height among 2397 European ancestry children in the Avon Longitudinal Study of Parents and Children (ALSPAC) cohort[28,29], as well as in predicting risk for 11 complex diseases amongst ~400,000 European ancestry participants in the UK Biobank[30]. An R toolkit implementing the method developed in this work, called FHPRS (Family History-assisted Polygenic Risk Score), is openly available at https://github.com/tianyuan-lu/PRS-FH-Prediction.

## Results

**Inference of under-captured genetic components in family history**. We propose a conceptual latent factor model to account for genetic components that are not modelled by polygenic risk scores but could be inferred from parental history (Methods). Briefly, for a continuous polygenic trait (Fig. 1a), we assume that its genetic determinants can be partitioned into two orthogonal genetic components: one component captured by a polygenic risk score used for prediction, and the other component representing under-captured genetic effects. The under-captured genetic component could include the effects of unmeasured common variants, rare variants, gene-by-environment interactions, epistasis, intra-familial shared environmental or lifestyle factors, etc. We suppose that these two genetic components are independently passed on from the parents to the children, hence parental measures of the trait may partially inform the under-captured genetic component (Methods and Fig. 1a). This model can be adapted for binary diseases, wherein the parental disease history may inform the underlying genetic liability (Fig. 1b).

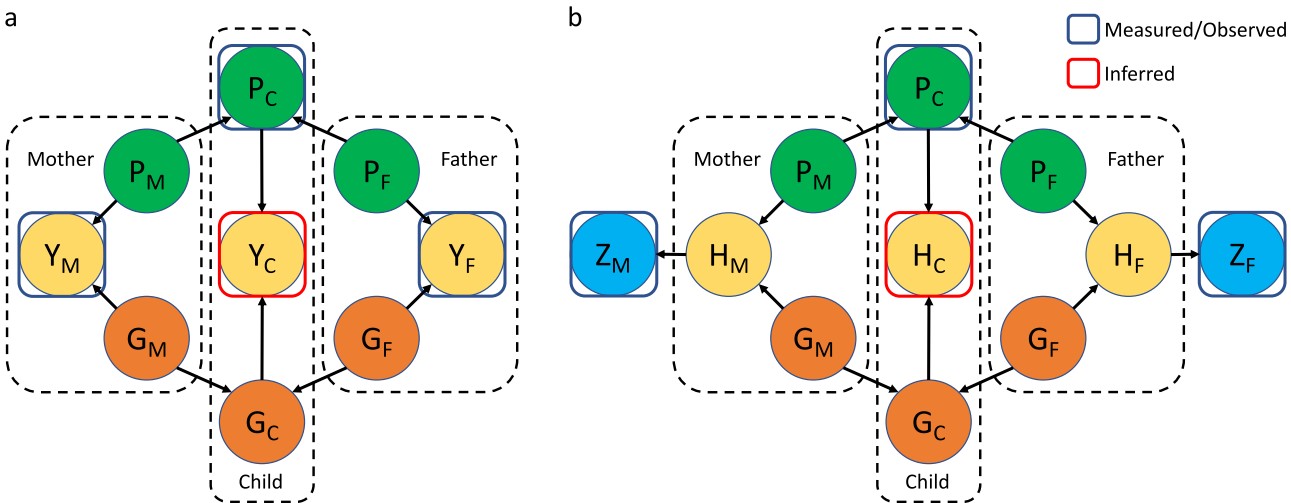

**Fig. 1 Causal diagrams representing latent factor models. a** For the parents and the children separately, a continuous trait $Y$ is determined by a polygenic component $P$ captured by a polygenic risk score, and a latent genetic component $G$ independent of $P$. **b** A binary trait $Z$ is determined by the underlying genetic liability $H$. Analogous to $Y$ in **a**, $H$ is jointly determined by $P$ and $G$. Prediction combining a polygenic risk score and parental information is equivalent to inference of the distribution of $Y$ in **a** and $H$ in **b** for the children. The two genetic components, $P$ and $G$, are assumed to be independently passed on from the parents to the offspring. Trait heritability is assumed to be constant across different generations.

Based on this model, a multivariate predictive model for a continuous trait or the genetic liability for a disease can be created by combining a polygenic risk score and family history, requiring only (1) the magnitude of association between a polygenic risk score and the target trait or disease, and (2) the magnitude of association between parents' trait measures or disease history and the target trait or disease among children (Methods). Importantly, these estimates can be obtained from separate well-powered reference cohort studies, without the need to access individual-level information for training a predictive model. Subsequently, predictions for individuals in a test population or patients in clinics can be obtained based on their genotypes and parental trait measures or family disease history.

**Improved height prediction among children by incorporating parental height measures**. We assessed the performance of this joint predictor in predicting children's adult height. A polygenic risk score for age and sex-adjusted adult height z-score was recently developed using resources from the UK Biobank and the Genetic Investigation of ANthropometric Traits consortium[7,30,31]. On an out-of-sample test dataset in the UK Biobank, this score explained 36.7% of the total variance in height z-score[7]. On the other hand, an observational study–the Erasmus Rucphen Family Study–found that the mid-parental height z-score was able to explain 44.9% of the total variance in offspring height z-scores[32]. Based on these estimates, we derived that 58.1% of the total height z-score variance was under-captured and could be partially inferred from parental height measures (Methods).

A joint predictor combining this polygenic risk score and mid-parental height was then obtained for European ancestry children in the ALSPAC cohort (Methods and Supplementary Table 1). Consequently, among 941 genotyped children who had both biological parents' height measures, this joint predictor was able to explain 55.3% of the total variance in their sex-adjusted adult height z-score (Fig. 2a). In contrast, similar to metrics obtained in the literature, the polygenic risk score alone explained 38.2% of the total variance, while the mid-parental height z-score alone explained 43.1% (Fig. 2a). As expected, the joint predictor also achieved the highest prediction accuracy with a root-mean-square error (RMSE) of 4.212 cm, compared to 4.955 cm by the polygenic risk score and 4.752 cm by the mid-parental height z-score (Fig. 2b).

Furthermore, the polygenic prediction could also be improved when only one parent's height was measured (Methods). Specifically, among 2246 children with maternal height measures, a joint predictor incorporating the polygenic risk score and maternal height z-score explained 47.1% of the total variance and achieved an RMSE of 4.584 cm (Fig. 2c, d). Among 1092 children with paternal height measures, a joint predictor incorporating the polygenic risk score and paternal height z-score explained 47.0% of the total variance and achieved an RMSE of 4.607 cm (Fig. 2e, f). Evidently, these joint predictors demonstrated superior predictive performance over the polygenic risk score or parental height alone.

Notably, all joint predictors demonstrated almost identical predictive performance as an in-sample combination of the polygenic risk score and parental height z-scores obtained from multivariate linear regression using individual-level data (Methods and Fig. 2).

**Improved complex disease risk prediction by incorporating parental disease history**. Next, we tested whether a joint predictor could improve polygenic risk prediction for 11 complex diseases in the UK Biobank (Supplementary Table 2). Polygenic risk scores have been recently developed for each of these diseases and are documented in the PGS Catalog (Supplementary Table 3)[33]. Estimation

of under-captured genetic effects was conducted based on a training dataset consisting of 10% of the participants from the UK Biobank (Methods and Supplementary Fig. 1). For most of these complex diseases, individuals having a parental disease history were significantly more likely to have the corresponding disease (Supplementary Table 4). The magnitudes of association reflect that a substantial proportion of genetic influence is not captured by the polygenic risk scores (Supplementary Table 4).

The incorporation of parental disease history into polygenic risk predictions led to a re-stratification of the predicted risks (Fig. 3a). Specifically, individuals with a parental disease history would be considered at an elevated level of risk compared to those with a similar polygenic risk score but without a parental disease history. The discriminative power of polygenic risk scores in identifying individuals who developed the corresponding diseases was significantly improved (Table 1 and Supplementary Table 5). For instance, combined with age, sex, genotyping array, recruitment centre, and the first 10 genetic principal components, a joint predictor for myocardial infarction achieved an area under the receiver operating characteristic curve (AUROC) of 0.7625 (Fig. 3b) and an area under the precision-recall curve (AUPRC) of 0.0834 (Fig. 3c), based on a test dataset consisting of 90% of the participants from the UK Biobank (Methods and Supplementary Fig. 1). In contrast, prediction based on the polygenic risk score (with other covariates but without parental history) had an AUROC of 0.7567 (DeLong's test $p$-value $= 5.3 \times 10^{-13}$) and an AUPRC of 0.0800; prediction based on the parental disease history (with other covariates but without the polygenic risk score) had an AUROC of 0.7375 (DeLong's test $p$-value $= 3.0 \times 10^{-126}$) and an AUPRC of 0.0671 (Fig. 3b, c).

Meanwhile, the joint predictor could more accurately assign individuals into higher-vs-lower risk groups, indicated by positive net-re-classification indices (NRI) across different percentile cut-offs and positive integrated discrimination improvement (IDI) indices (Table 1). For example, at the 95th percentile cut-off (i.e. 5% of the population to be considered at high risk), the joint predictor for Alzheimer's disease had the highest NRI of 5.76% over the polygenic risk score for Alzheimer's disease, followed by the joint predictor for chronic obstructive pulmonary disease (COPD), with an NRI of 5.07% (Table 1). Notably, the joint predictor for Alzheimer's disease was significantly associated with carrying one or two *APOE* e4 alleles (Methods and Supplementary Fig. 2a and b), which is a well-known genetic risk factor for Alzheimer's disease[34–36] but is not included in the polygenic risk score for this analysis. In addition, individuals with a higher score in the joint predictor for COPD were slightly yet significantly more likely to be ever-smokers, while the polygenic risk score alone did not capture the genetic predisposition to smoking (Methods and Supplementary Fig. 2c).

For most diseases under investigation, the joint predictors consistently achieved comparable predictive performance to gold-standard predictors obtained by fitting multivariate logistic regression models based on individual-level data from the training datasets (Methods and Supplementary Fig. 1). For breast cancer, lung cancer, stroke, and Alzheimer's disease, the joint predictors even achieved a marginally higher AUROC than the data-driven predictors (Table 1 and Supplementary Fig. 3), although the joint predictors for ischemic heart disease and type 2 diabetes did not appear to be ideal (Table 1 and Supplementary Fig. 3). However, as expected, all of the individual-level data-driven predictors demonstrated sensitivity to sample size (Fig. 3d and Supplementary Fig. 3). For instance, for myocardial infarction, if a regression-based predictor were to be derived using ≤20% of the individuals included in the current training dataset, its discriminative power would likely be worse than the polygenic risk score alone (Fig. 3d).

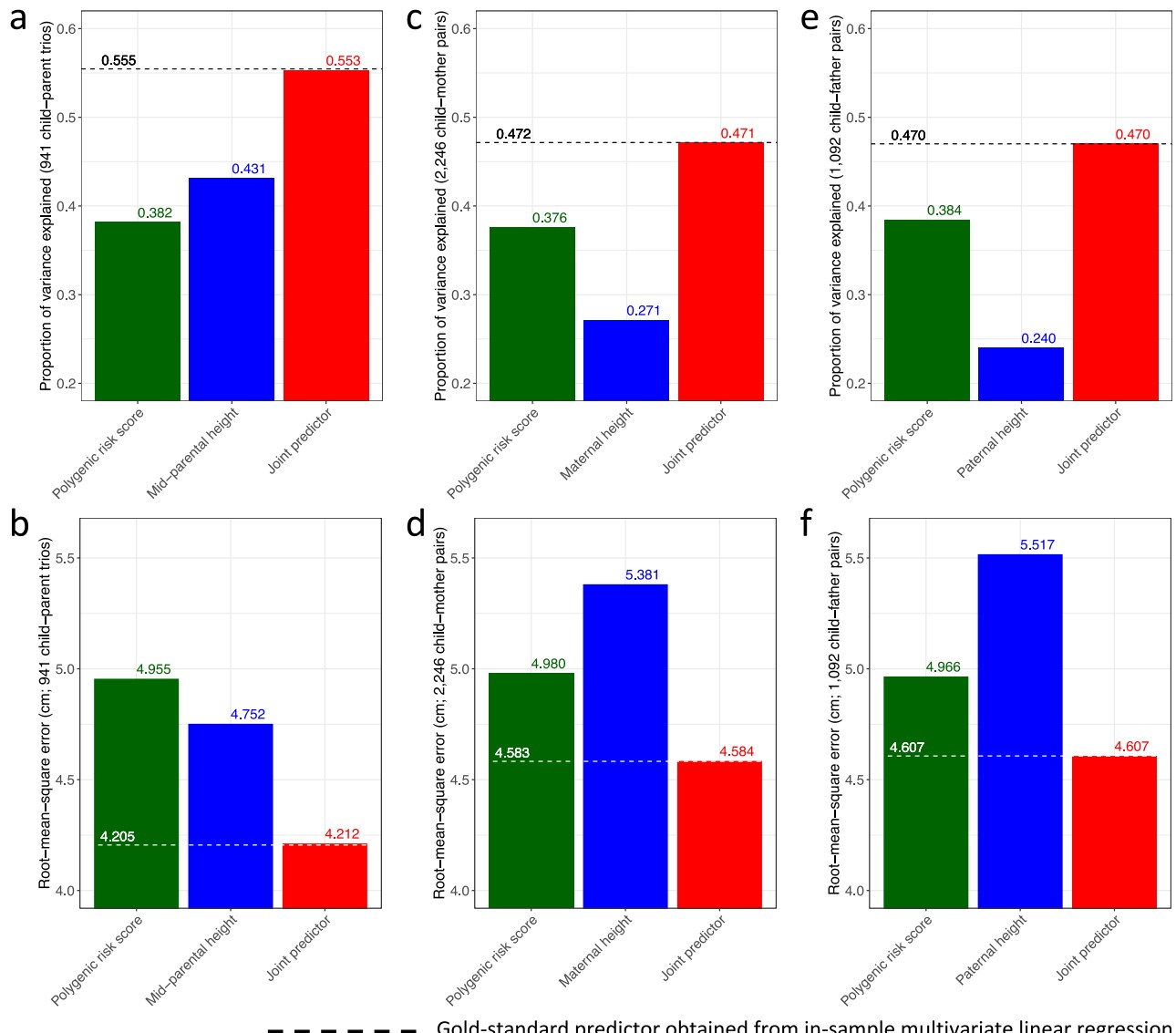

**Fig. 2 Comparison of predictive performance in predicting children's sex-adjusted adult height z-scores for a polygenic risk score, parental height measures, and a joint predictor.** A joint predictor combining the polygenic risk score and the mid-parental height predictor achieves **a** the highest proportion of variance explained, and **b** the lowest root-mean-square error based on 941 child-parent trios. A joint predictor incorporating **c**, **d** only the maternal height for 2246 child-mother pairs, or **e**, **f** only the paternal height for 1092 child-father pairs also outperforms the polygenic risk score and the single parental height predictor alone. These joint predictors have similar predictive performance as the corresponding gold-standard in-sample best linear unbiased predictors based on individual-level data, indicated by dashed lines.

## Discussion

Polygenic risk scores for complex traits are effective research tools and may greatly improve personalized health care in clinical practice[1,2]. The development of polygenic risk scores relies on the accurate characterization of genetic determinants in large-scale GWASs. Due to the limitation of resources, most existing polygenic risk scores are restricted to modelling linear additive effects of common genetic variants, thus may benefit from being combined with predictors that are able to capture more elusive genetic effects. In this study, we have proposed a simple latent factor model to quantify and extract heritability not captured by polygenic risk scores but inferable from family history. We have systematically investigated the utility of adding family history into polygenic risk score-based polygenic risk prediction models.

The combination of parental height measures and a polygenic risk score for height brought substantial improvements in adult height prediction. In fact, the proportion of variance explained in adult height z-scores by our joint predictor in the ALSPAC cohort was close to the estimated total SNP heritability of height z-scores[37–40]. On the other hand, for 11 complex diseases, the joint predictors consistently demonstrated stronger discriminative power in identifying individuals at an elevated level of risk, although the improvements appeared to be marginal. This may reflect the limited sensitivity of relevant metrics to the addition of new predictors[41], as we observed in simulation studies (Supplementary Note 1 and Supplementary Fig. 4). Nonetheless, by jointly examining all metrics (AUROC, AUPRC, NRI, and IDI) as well as the distribution of predicted risks, we posit that the resulting risk re-stratification could still benefit up to thousands of individuals at the biobank scale.

In developing this prediction scheme, we partitioned trait or disease heritability onto two orthogonal genetic components, where family history is assumed to partially inform a latent genetic component despite being correlated with the polygenic

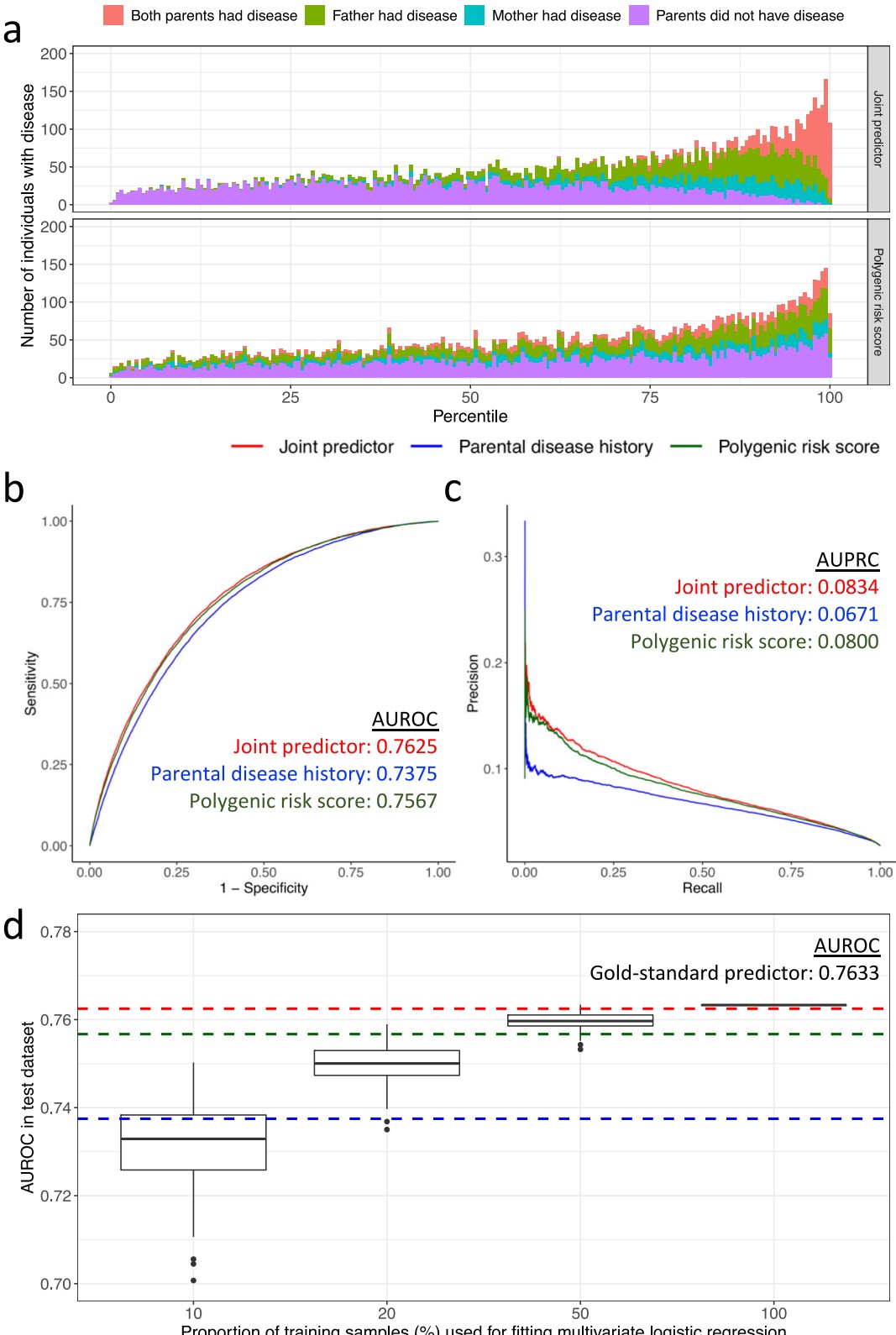

**Fig. 3 A joint predictor improves risk stratification for myocardial infarction in the UK Biobank. a** Incorporation of parental disease history leads to calibration of polygenic risk scores. Individuals with a parental disease history are more likely to be considered at risk. The joint predictor achieves higher **b** area under the receiver operating characteristic curve (AUROC) and **c** area under the precision-recall curve (AUPRC), compared to the polygenic risk score and the parental disease history. **d** Performance of a data-driven joint predictor obtained from multivariate logistic regression is sensitive to sample size when using individual-level data. Boxplots represent distributions of AUROC obtained in 100 replicates corresponding to different sample sizes. AUROC of the joint predictor, polygenic risk score, and parental disease history are indicated by red, green, and blue dashed lines, respectively.

**Table 1 Improved discriminative power of a joint predictor in identifying individuals at an elevated risk of disease based on the UK Biobank test dataset.**

| Disease | AUROC[a] (p-value of DeLong's test[b]) | | | NRI[c] (%) at score percentile cut-off | | | | IDI[c] (%) |
|---|---|---|---|---|---|---|---|---|
| | Polygenic risk score | Parental history | Joint predictor | 50 | 80 | 95 | 99 | |
| Breast cancer | 0.6443 ($5.8 \times 10^{-16}$) | 0.5771 ($8.9 \times 10^{-213}$) | 0.6500 | +0.94 | +0.84 | +0.14 | +0.17 | +0.56 |
| Prostate cancer | 0.7567 ($1.4 \times 10^{-3}$) | 0.7071 ($2.6 \times 10^{-197}$) | 0.7588 | +0.29 | +0.24 | +0.68 | +0.12 | +0.21 |
| Colorectal cancer | 0.6755 ($5.4 \times 10^{-8}$) | 0.6676 ($2.1 \times 10^{-41}$) | 0.6768 | +0.27 | +0.64 | +0.45 | 0 | +0.13 |
| Lung cancer | 0.6915 ($1.6 \times 10^{-7}$) | 0.6947 ($1.3 \times 10^{-6}$) | 0.6975 | +2.46 | +5.07 | +3.01 | +1.12 | +2.60 |
| Myocardial infarction | 0.7567 ($5.3 \times 10^{-13}$) | 0.7375 ($3.0 \times 10^{-126}$) | 0.7625 | +0.85 | +0.70 | +0.96 | +0.09 | +0.58 |
| Ischemic heart disease | 0.7445 ($1.5 \times 10^{-38}$) | 0.7398 ($3.4 \times 10^{-297}$) | 0.7520 | +1.10 | +0.96 | +0.57 | +0.22 | +0.75 |
| Stroke | 0.6862 ($6.2 \times 10^{-10}$) | 0.6802 ($8.7 \times 10^{-16}$) | 0.6870 | +0.41 | +0.06 | +0.38 | +0.14 | +0.49 |
| Type 2 diabetes | 0.7129 ($6.7 \times 10^{-31}$) | 0.7072 ($<1.0 \times 10^{-350}$) | 0.7266 | +1.71 | +2.50 | +1.47 | +0.89 | +1.37 |
| Alzheimer's disease | 0.7571 ($1.4 \times 10^{-7}$) | 0.7616 ($1.5 \times 10^{-3}$) | 0.7645 | +4.48 | +6.65 | +5.76 | +2.18 | +4.02 |
| Parkinson's disease | 0.7482 ($2.6 \times 10^{-2}$) | 0.7412 ($2.1 \times 10^{-13}$) | 0.7507 | +1.86 | +2.21 | +0.59 | +1.08 | +1.22 |
| COPD | 0.7325 ($4.4 \times 10^{-58}$) | 0.7448 ($2.0 \times 10^{-11}$) | 0.7463 | +7.03 | +11.88 | +5.07 | +1.60 | +5.96 |

[a]Including effects of age, sex (except for breast cancer and prostate cancer), genotyping array, recruitment centre, and the first 10 genetic principal components.
[b]Comparing AUROC of the polygenic risk score or parental history to that of the joint predictor.
[c]Comparing the joint predictors to the corresponding polygenic risk scores; a positive number indicates improved re-classification.

risk score. To account for the polygenicity and complexity of the underlying genetic effects, the genetic components were assumed to be normally distributed. We recognize that while these assumptions have strength in facilitating model specification and construction of joint predictors, they may not be theoretically optimal. For example, if strong gene-by-environment interaction effects exist, the under-captured genetic component should, at least in part, be correlated with the polygenic risk score. However, correlation or interaction between the two genetic components is not identifiable without accessing individual-level training data. Encouragingly, our simulation studies indicated that our method could tolerate mild-to-moderate violations of model assumptions as well as errors in estimating model parameters (Supplementary Notes 2 and 3, and Supplementary Figs. 5–10). Moreover, despite these strong assumptions, we found in practice that polygenic prediction could consistently be improved through our models built with this point of view, regardless of the underlying genetic architecture of complex traits, disease prevalence, and the methods adopted to develop polygenic risk scores[33]. In contrast to multivariate regression-based joint predictors, our method can leverage results of association tests obtained from separate cohort studies with high statistical power, thus this method does not suffer when large individual-level training datasets are unavailable.

Our findings have important implications for developing polygenic risk predictors, since the under-captured genetic components may have known sources. For instance, a large proportion of Alzheimer's disease risk heritability is conferred by the common allele of *APOE* e4[34–36]. Not surprisingly, the parental disease history of Alzheimer's disease appeared to be more predictive of the disease risk compared to a polygenic risk score not including this allele, while the joint predictor demonstrated clear advantages in risk stratification. Furthermore, by design, inherited risk factor exposure was also included in the under-captured genetic component. Smoking is one of the most important risk factors for COPD[42] and is heritable[43,44]. In our model, a family disease history of COPD may partially capture the genetic predisposition to smoking as well as other risk factors that are not fully represented in the polygenic risk score. As a result, the corresponding joint predictor aggregating additional genetic risks demonstrated a prominent improvement over the polygenic risk score. These results not only support the utility of family history in enhancing predictive power but also encourage explicit modelling of large effects such as monogenic causes or significant intra-familial shared risk factor exposures in polygenic risk prediction.

We note possible model mis-specification for ischemic heart disease and type 2 diabetes because the joint predictors we constructed displayed compromised predictive performance compared to the gold-standard predictors. We hypothesize that this may be due to errors in the definitions of disease in the parental histories as well as in the medical histories of the participants. Specifically, sub-classifications of diseases were not available for parental heart disease and diabetes, where the former may include various types of diseases affecting the cardiovascular system, and the latter may include both type 1 and type 2 diabetes. Consistent disease definitions and comprehensive phenotyping are thus required for validating our findings in clinical practice and in research.

We anticipate that the predictive performance of the joint predictors could be further enhanced with additional knowledge of family history, such as the disease history of relatives other than the parents, using empirical genetic relatedness based on pedigree information. However, we expect that information gained from second-degree or more distant relatives would be less significant compared to first-degree relatives. Furthermore, in the UK Biobank, the parental disease history largely reflected the parents' lifetime risk given that the participants (children) were aged above 40 years upon recruitment. Hence, appropriate modelling of age-dependent risks should be pursued for most complex diseases that do not have an early onset, if the disease history of younger relatives were to be considered.

Last, our findings should be considered specific to the study populations. Participants in our study cohorts are predominantly of European ancestries, yet it has been widely recognized that polygenic risk scores can have largely attenuated predictive performance when applied to populations of different genetic ancestries[45,46]. Hence, we recommend extensive validations of our results in diverse populations as well as the development of ancestry-specific polygenic risk scores. Adding to this restriction, we also note that participants in the UK Biobank have been shown to be slightly healthier, less obese, and less likely to smoke and drink alcohol than the general population in the United Kingdom[47]. In particular, the average age of the study cohort was substantially younger than the average age of onset for Alzheimer's disease and Parkinson's disease[48,49], thus the corresponding disease prevalence was low. Therefore, if a joint predictor were to be constructed for other populations, we strongly recommend estimating model parameters based on reference cohort studies with similar demographic characteristics and prevalence of the disease. This may be particularly important for large cohort

studies and population-level screening programs where family history information is less comprehensive than in clinical settings.

In summary, we have developed a risk calculation scheme by incorporating family history into polygenic risk scores. We found a substantial proportion of complex trait heritability was under-captured and could be partly inferred from family history. Our findings support the utility of combining family history with polygenic risk scores as well as investigations into complex genetic effects not captured by existing polygenic risk scores for improved genetic risk prediction.

## Methods

**Related methods.** Combining a genetic risk score with measures of traits or disease status of relatives is not unprecedented. The most straightforward approach is to fit a multivariate regression model including both the genetic risk score and family history variables as predictors[7,25]. Alternatively, extensions of the best linear unbiased prediction (BLUP) method have been proposed by appending an empirical relatedness matrix based on pedigree information to a genotype-based relatedness matrix for modelling random effects in a mixed model setting[26,27]. Using a similar framework, family history may also improve the power of genetic association tests[19]. However, these methods require access to a training dataset that simultaneously contains phenotypes, genotypes, and family history information. For modelling disease outcomes, a large number of cases is needed to ensure statistical power. This is often unlikely due to confidentiality restrictions or logistical constraints. In addition, the variance-covariance structure of genetic components specified in our method is also similar to that implemented in a few methods using family disease status to modify genetic risk estimates in known risk loci[50–52]. Nonetheless, these methods require a pre-specified estimate of disease heritability, which may be prone to error if no reference for the targeted population is available or if unmeasured covariate effects are not properly accounted for.

**A latent factor model for polygenic inheritance in parent-child trios.** We first consider a simple latent factor model for a normally distributed polygenic trait

$$Y_M = \alpha P_M + \beta G_M + \epsilon_M \tag{1}$$

$$Y_F = \alpha P_F + \beta G_F + \epsilon_F \tag{2}$$

$$Y_C = \alpha P_C + \beta G_C + \epsilon_C \tag{3}$$

where the subscripts $M$, $F$, and $C$ stand for the populations of mothers, fathers, and children, respectively. The mothers and the fathers are assumed to be unrelated. $P$ represents the genetic component already captured by a polygenic risk score, while $G$ represents the under-captured genetic component, including under-captured linear additive effects, non-additive effects, rare pathogenic variant effects, gene-gene or gene-environment interaction effects, inherited exposure to risk factors, etc. Importantly, $G$ is assumed to be independent of $P$, and these two components are independently passed on from the parents to the children. We realize that this may be an unrealistic assumption, particularly for common effects that were not captured in existing polygenic risk scores, but this assumption enables the conceptual development of our latent model. Results in Fig. 2 and Table 1 justify the usefulness of the perspective. $\epsilon$ captures the non-genetic residual variance in $Y$, where $\epsilon \perp (P, G)$. We assume both $P$ and $G$ are normally distributed because $P$, under a polygenic model, consists of multiple independent effects, each having an infinitesimal effect size, while $G$ has a multifactorial nature. Without loss of generality, we suppose that $Y$, $P$, and $G$ are scaled to have zero mean and unit variance.

We further assume that the effects of the modelled polygenic component and the under-captured genetic component, $\alpha$ and $\beta$, are invariant in the populations of parents and children from the same study. Naturally, the overall heritability of this trait is $\alpha^2 + \beta^2$.

Next, we describe the joint distribution of $P$, $G$, and $Y$, where $P = (P_M, P_F, P_C)^\top$, $G = (G_M, G_F, G_C)^\top$, and $Y = (Y_M, Y_F, Y_C)^\top$:

$$\begin{pmatrix} P \\ G \\ Y \end{pmatrix} \sim \mathcal{N}\left( \begin{pmatrix} 0 \\ 0 \\ 0 \end{pmatrix}, \begin{pmatrix} \Sigma_{PP} & \Sigma_{PG} & \Sigma_{PY} \\ \Sigma_{GP} & \Sigma_{GG} & \Sigma_{GY} \\ \Sigma_{YP} & \Sigma_{YG} & \Sigma_{YY} \end{pmatrix} \right) \tag{4}$$

Based on the polygenic model assumptions, the covariance between the children's genetic components and their parents' corresponding genetic components is expected to be $\frac{1}{2}$, as each child inherits half of the trait-determining alleles from their mother and the other half from their father. Hence, we specify empirical covariance matrices for $\Sigma_{PP}$ and $\Sigma_{GG}$:

$$\Sigma_{PP} = \Sigma_{GG} = \begin{pmatrix} 1 & 0 & \frac{1}{2} \\ 0 & 1 & \frac{1}{2} \\ \frac{1}{2} & \frac{1}{2} & 1 \end{pmatrix} \tag{5}$$

It is noteworthy that by specifying these variance-covariance matrices, we assume no consanguinity and no assortative mating.

Since $P$ and $G$ are assumed to be independent of each other, we set:

$$\Sigma_{PG} = \Sigma_{GP} = \mathbf{0} \tag{6}$$

For associations between the trait and genetic components, we can derive that $Cov(P_k, Y_k) = Cov(P_k, \alpha P_k + \beta G_k + \epsilon_k) = \alpha$ for $k \in \{M, F, C\}$ and $Cov(P_C, Y_{M/F}) = \mathrm{Cov}(P_C, \alpha P_{M/F} + \beta G_{M/F} + \epsilon_{M/F}) = \frac{\alpha}{2}$ (the subscript $M/F$ represents maternal or paternal component), thus

$$\Sigma_{PY} = \Sigma_{YP} = \begin{pmatrix} \alpha & 0 & \frac{\alpha}{2} \\ 0 & \alpha & \frac{\alpha}{2} \\ \frac{\alpha}{2} & \frac{\alpha}{2} & \alpha \end{pmatrix} \tag{7}$$

Similarly,

$$\Sigma_{GY} = \Sigma_{YG} = \begin{pmatrix} \beta & 0 & \frac{\beta}{2} \\ 0 & \beta & \frac{\beta}{2} \\ \frac{\beta}{2} & \frac{\beta}{2} & \beta \end{pmatrix} \tag{8}$$

Furthermore, the covariance between the children's trait and the parents' trait $\mathrm{Cov}(Y_C, Y_{M/F}) = \mathrm{Cov}(\alpha P_C + \beta G_C + \epsilon_C, \alpha P_{M/F} + \beta G_{M/F} + \epsilon_{M/F}) = \frac{\alpha^2 + \beta^2}{2}$. Therefore,

$$\Sigma_{YY} = \begin{pmatrix} 1 & 0 & \frac{\alpha^2 + \beta^2}{2} \\ 0 & 1 & \frac{\alpha^2 + \beta^2}{2} \\ \frac{\alpha^2 + \beta^2}{2} & \frac{\alpha^2 + \beta^2}{2} & 1 \end{pmatrix} \tag{9}$$

Equation (9) implies that if a combination of the parental trait measures, usually the mid-parental trait measure, is used to predict the children's trait, the expected proportion of variance explained is

$$Corr^2\left(Y_C, \frac{Y_M + Y_F}{2}\right) = \left( \frac{\frac{1}{2}\mathrm{Cov}(Y_C, Y_M) + \frac{1}{2}\mathrm{Cov}(Y_C, Y_F)}{\sqrt{\mathrm{Var}(Y_C)\mathrm{Var}(\frac{Y_M + Y_F}{2})}} \right)^2 = \frac{(\alpha^2 + \beta^2)^2}{2} \tag{10}$$

In practice, an estimate of $\alpha$ can be obtained from studies developing polygenic risk scores, i.e. $\hat{\alpha}^2 = \widehat{Corr}^2(P, Y)$. Meanwhile, observational studies reporting the magnitude of associations between parental trait measures and the children's trait measures can inform the under-captured heritability $\beta^2$, e.g.

$$\hat{\beta}^2 = \sqrt{2\widehat{Corr}^2\left(Y_C, \frac{Y_M + Y_F}{2}\right)} - \hat{\alpha}^2 \tag{11}$$

**Continuous trait prediction incorporating parental trait measures.** Based on the distributional assumptions in Eq. (4), we are able to infer $Y_C$ with the parental trait measures ($Y_M$ and/or $Y_F$) and the children's polygenic risk score ($P_C$), if we have estimates of $\alpha$ and $\beta$. Specifically, given the properties of the multivariate normal distribution,

$$\begin{pmatrix} Y_C \\ Y_M \\ Y_F \\ P_C \end{pmatrix} \sim \mathcal{N}\left( \begin{pmatrix} 0 \\ 0 \\ 0 \\ 0 \end{pmatrix}, \begin{pmatrix} 1 & \frac{\alpha^2 + \beta^2}{2} & \frac{\alpha^2 + \beta^2}{2} & \alpha \\ \frac{\alpha^2 + \beta^2}{2} & 1 & 0 & \frac{\alpha}{2} \\ \frac{\alpha^2 + \beta^2}{2} & 0 & 1 & \frac{\alpha}{2} \\ \alpha & \frac{\alpha}{2} & \frac{\alpha}{2} & 1 \end{pmatrix} \right) \tag{12}$$

For the $j$th individual with a polygenic risk score $p_{Cj}$ and parental measures of the trait $y_{Mj}, y_{Fj}$, we use the conditional expectation as the predictor for the trait:

$$\bar{Y}_{Cj} = E(Y_C | Y_M = y_{Mj}, Y_F = y_{Fj}, P_C = p_{Cj}) = \begin{pmatrix} \frac{\hat{\alpha}^2 + \hat{\beta}^2}{2} & \frac{\hat{\alpha}^2 + \hat{\beta}^2}{2} & \hat{\alpha} \end{pmatrix} \begin{pmatrix} 1 & 0 & \frac{\hat{\alpha}}{2} \\ 0 & 1 & \frac{\hat{\alpha}}{2} \\ \frac{\hat{\alpha}}{2} & \frac{\hat{\alpha}}{2} & 1 \end{pmatrix}^{-1} \begin{pmatrix} y_{Mj} \\ y_{Fj} \\ p_{Cj} \end{pmatrix} \tag{13}$$

Alternatively, we can create a predictor using the polygenic risk score with one parental measure if only the maternal or the paternal measure is available:

$$\bar{Y}_{Cj} = E(Y_C | Y_{M/F} = y_{M/Fj}, P_C = p_{Cj}) = \begin{pmatrix} \frac{\hat{\alpha}^2 + \hat{\beta}^2}{2} & \hat{\alpha} \end{pmatrix} \begin{pmatrix} 1 & \frac{\hat{\alpha}}{2} \\ \frac{\hat{\alpha}}{2} & 1 \end{pmatrix}^{-1} \begin{pmatrix} y_{M/Fj} \\ p_{Cj} \end{pmatrix} \tag{14}$$

This framework is generalizable to any degree of family relationships, with modification of the variance-covariance matrices, possibly using empirical estimates of genetic relatedness. However, because information about more distantly related family members is more difficult to obtain, and is rarely complete in large cohort studies, and because parental information is the most relevant to risk prediction, we focus on parental trait measures or disease history in this work.

**Modelling latent genetic components for binary diseases.** For binary diseases, we combine a liability model with the above latent factor model and explicitly model measurable covariate effects (such as age and sex) since they are non-trivial

in this case. We denote the mothers', fathers' and children's diseases of interest as $Z_M$, $Z_F$, and $Z_C$, respectively. We assume that

$$(Z_k | P_k = p, G_k = g, Q_k = \mathbf{q}) \sim \text{Bernoulli}\left( \frac{e^{\mu_0 + \alpha p + \beta g + \gamma \mathbf{q}}}{1 + e^{\mu_0 + \alpha p + \beta g + \gamma \mathbf{q}}} \right) \quad (15)$$

for $k \in \{M, F, C\}$, where $Q$ represents measured (single or multiple) covariates with effects $\gamma$, and $\mu_0$ denotes the baseline odds (on the logarithmic scale) of the disease in the target population. $P$ and $G$ are assumed to be independently distributed with $P \sim \mathcal{N}(0, 1)$ and $G \sim \mathcal{N}(0, 1)$.

If we introduce $H = \alpha P + \beta G$, where ($H \sim \mathcal{N}(0, \alpha^2 + \beta^2)$) includes both genetic components, then Eq. (15) simplifies to

$$(Z_k | H_k = h, Q_k = \mathbf{q}) \sim \text{Bernoulli}\left( \frac{e^{\mu_0 + h + \gamma \mathbf{q}}}{1 + e^{\mu_0 + h + \gamma \mathbf{q}}} \right) \quad (16)$$

With the Bernoulli distribution assumptions, we adopt a logit-link in modelling the genetic liability, instead of the probit-link which is more commonly used for liability threshold models[19]. This choice facilitates the estimation of the model parameters from the results of widely-implemented logistic regression models. Nevertheless, heritability estimates can still be obtained. As shown previously[53], an approximation to obtain the effect size estimate on the liability scale is

$$\tau_\alpha \approx \frac{\Phi^{-1}[F_{\text{Logistic}}(\mu_0 + \alpha)] - \Phi^{-1}[F_{\text{Logistic}}(\mu_0)]}{\sqrt{1 + \{\Phi^{-1}[F_{\text{Logistic}}(\mu_0 + \alpha)] - \Phi^{-1}[F_{\text{Logistic}}(\mu_0)]\}^2 \text{Var}(P)}} \quad (17)$$

and

$$\tau_\beta \approx \frac{\Phi^{-1}[F_{\text{Logistic}}(\mu_0 + \beta)] - \Phi^{-1}[F_{\text{Logistic}}(\mu_0)]}{\sqrt{1 + \{\Phi^{-1}[F_{\text{Logistic}}(\mu_0 + \beta)] - \Phi^{-1}[F_{\text{Logistic}}(\mu_0)]\}^2 \text{Var}(G)}} \quad (18)$$

where $\Phi$ is the standard normal cumulative distribution function, $F_{\text{Logistic}}$ is the cumulative distribution function of logistic distribution with a mean of 0 and variance of $\frac{\pi^2}{3}$, i.e. $F_{\text{Logistic}}(x) = \frac{1}{1 + \exp\{-\frac{\pi}{\sqrt{3}} x\}}$, and $\text{Var}(P) = \text{Var}(G) = 1$.

Consequently, heritability captured by the polygenic risk score can be estimated as $\tau_\alpha^2$, while the under-captured heritability can be estimated as $\tau_\beta^2$. Notably, unlike the model for continuous traits (Section A latent factor model for polygenic inheritance in parent-child trios) where $\alpha$ and $\beta$ are automatically bounded ($0 \le \alpha^2 + \beta^2 \le 1$), here, the effects of the genetic components $P$ and $G$ are unconstrained.

Estimates of the unknown parameters can be obtained from summary statistics of existing observational studies. That is, we can directly obtain $\hat{\alpha}$ based on the magnitude of association between a polygenic risk score and the disease risk, i.e. odds ratio (OR) per one standard deviation increase in the polygenic risk score, together with $\hat{\mu}_0$ and $\hat{\gamma}$. Furthermore, we can empirically estimate $\beta$ if we have an estimate of the association between a parental disease history and the disease risk amongst the children.

Specifically, from the joint distribution

$$f(Z_C, Z_{M/F}) = \int_{H_C, H_{M/F}} f(Z_C, Z_{M/F}, H_C, H_{M/F}) \\ = \int_{H_C, H_{M/F}} f(Z_C | H_C) f(Z_{M/F} | H_{M/F}) f(H_C, H_{M/F}) \quad (19)$$

where

$$f(Z_k | H_k = h) = \left( \frac{e^{\mu_0 + h + \gamma \mathbf{q}}}{1 + e^{\mu_0 + h + \gamma \mathbf{q}}} \right)^{z_k} \left( \frac{1}{1 + e^{\mu_0 + h + \gamma \mathbf{q}}} \right)^{1 - z_k} \quad (20)$$

for $k \in \{M, F, C\}$, and

$$\begin{pmatrix} H_C \\ H_{M/F} \end{pmatrix} \sim \mathcal{N}\left( \begin{pmatrix} 0 \\ 0 \end{pmatrix}, \begin{pmatrix} \alpha^2 + \beta^2 & \frac{\alpha^2 + \beta^2}{2} \\ \frac{\alpha^2 + \beta^2}{2} & \alpha^2 + \beta^2 \end{pmatrix} \right) \quad (21)$$

we have an explicit expression for the OR based on either the maternal or the paternal disease history as a function of the unknown $\beta$:

$$\widetilde{\text{OR}}_{M/F} = \frac{f(Z_C = 1, Z_{M/F} = 1) / f(Z_C = 0, Z_{M/F} = 1)}{f(Z_C = 1, Z_{M/F} = 0) / f(Z_C = 0, Z_{M/F} = 0)} \quad (22)$$

Therefore, we perform a numerical line search to obtain an empirical estimate of $\beta$ such that the theoretical $\widetilde{\text{OR}}_{M/F}$ is close to the observed $\widehat{\text{OR}}_{M/F}$:

$$\hat{\beta} = \underset{\beta}{argmin}\{|(\log(\widehat{\text{OR}}_M) - \log(\widetilde{\text{OR}}_M)) + (\log(\widehat{\text{OR}}_F) - \log(\widetilde{\text{OR}}_F))|\} \quad (23)$$

It should be noted that the OR associated with the maternal disease history may differ from that based on the paternal disease history because sex is included as a covariate (in $Q$) in this model.

## Binary disease risk prediction incorporating parental disease history.
Similar to predicting a continuous trait (Section Continuous trait prediction incorporating parental trait measures), we aim to infer $f(H_C | Z_M, Z_F, P_C)$. since the distribution of $H_C$ naturally informs $f(Z_C)$. Then we can use the conditional expectation as the predictor.

We observe that

$$f(H_C | Z_M, Z_F, P_C) = f(H_C | H_M, H_F, P_C) f(H_M | Z_M) f(H_F | Z_F) \quad (24)$$

has no closed-form solution for its expectation, since

$$\begin{pmatrix} H_C \\ H_M \\ H_F \\ P_C \end{pmatrix} \sim \mathcal{N}\left( \begin{pmatrix} 0 \\ 0 \\ 0 \\ 0 \end{pmatrix}, \begin{pmatrix} \alpha^2 + \beta^2 & \frac{\alpha^2 + \beta^2}{2} & \frac{\alpha^2 + \beta^2}{2} & \alpha \\ \frac{\alpha^2 + \beta^2}{2} & \alpha^2 + \beta^2 & 0 & \frac{\alpha}{2} \\ \frac{\alpha^2 + \beta^2}{2} & 0 & \alpha^2 + \beta^2 & \frac{\alpha}{2} \\ \alpha & \frac{\alpha}{2} & \frac{\alpha}{2} & 1 \end{pmatrix} \right) \quad (25)$$

and

$$f(H_k | Z_k = z_k) = \frac{f(Z_k = z_k | H_k) f(H_k)}{f(Z_k = z_k)} \propto \left( \frac{e^{\mu_0 + h_k + \gamma \mathbf{q}}}{1 + e^{\mu_0 + h_k + \gamma \mathbf{q}}} \right)^{z_k} \left( \frac{1}{1 + e^{\mu_0 + h_k + \gamma \mathbf{q}}} \right)^{(1 - z_k)} \left( e^{-\frac{h_k^2}{2(\alpha^2 + \beta^2)}} \right) \quad (26)$$

for $k \in \{M, F, C\}$. Therefore, we implement an importance sampling scheme to approximate this conditional distribution and derive its expectation:

For the $j$th individual with a polygenic risk score $p_{C,j}$ and parental disease history $z_{M,j} \in \{0, 1\}$, $z_{F,j} \in \{0, 1\}$, we randomly generate $L$ (a large number, e.g. 1,000,000) samples of $H_{M,j}$ and $H_{F,j}$ based on $f(H_M | Z_M = z_{M,j})$ and $f(H_F | Z_F = z_{F,j})$ in Eq. (26), respectively;

For the $l$-th sample of $H_{M,j}$ and $H_{F,j}$, denoted as $h_{M,j,l}$ and $h_{F,j,l}$, we derive

$$E_{(l)}(H_C | H_M = h_{M,j,l}, H_F = h_{F,j,l}, P_C = p_{C,j}) = \begin{pmatrix} \frac{\hat{\alpha}^2 + \hat{\beta}^2}{2} & \frac{\hat{\alpha}^2 + \hat{\beta}^2}{2} & \hat{\alpha} \end{pmatrix} \begin{pmatrix} \hat{\alpha}^2 + \hat{\beta}^2 & 0 & \frac{\hat{\alpha}}{2} \\ 0 & \hat{\alpha}^2 + \hat{\beta}^2 & \frac{\hat{\alpha}}{2} \\ \frac{\hat{\alpha}}{2} & \frac{\hat{\alpha}}{2} & 1 \end{pmatrix}^{-1} \begin{pmatrix} h_{M,j,l} \\ h_{F,j,l} \\ p_{C,j} \end{pmatrix} \quad (27)$$

We repeat for all $L$ samples and obtain the predictor for the $j$th individual as

$$\bar{H}_{C,j} = \frac{1}{L} \sum_{l=1}^{L} E_{(l)}(H_C | H_M = h_{M,j,l}, H_F = h_{F,j,l}, P_C = p_{C,j}) \quad (28)$$

Alternatively, we can utilize the disease history of only one parent, particularly when a disease is highly sex-specific, where Eq. (27) is modified as

$$E_{(l)}(H_C | H_{M/F} = h_{M/F,j,l}, P_C = p_{C,j}) = \begin{pmatrix} \frac{\hat{\alpha}^2 + \hat{\beta}^2}{2} & \hat{\alpha} \end{pmatrix} \begin{pmatrix} \hat{\alpha}^2 + \hat{\beta}^2 & \frac{\hat{\alpha}}{2} \\ \frac{\hat{\alpha}}{2} & 1 \end{pmatrix}^{-1} \begin{pmatrix} h_{M/F,j,l} \\ p_{C,j} \end{pmatrix} \quad (29)$$

and Eq. (28) is modified correspondingly as

$$\bar{H}_{C,j} = \frac{1}{L} \sum_{l=1}^{L} E_{(l)}(H_C | H_{M/F} = h_{M/F,j,l}, P_C = p_{C,j}) \quad (30)$$

## Predicting adult height for children in the Avon Longitudinal Study of Parents and Children.
From 1991 to 1992, the ALSPAC cohort recruited 14,541 pregnancies in the Bristol and Avon areas in the United Kingdom[28,29]. In addition, 913 pregnancies were enrolled in later phases of the study. The total sample size for analyses using data collected after the age of seven is therefore 15,454 pregnancies, resulting in 15,589 foetuses. Of these, 14,901 were alive at 1 year of age. Ethical approval for the study was obtained from the ALSPAC Ethics and Law Committee and the Local Research Ethics Committees. Consent for biological samples has been collected in accordance with the Human Tissue Act (2004). Informed consent for the use of data collected via questionnaires and clinics was obtained from participants following the recommendations of the ALSPAC Ethics and Law Committee at the time.

Genotyping of the children initially recruited into this cohort was conducted using the Illumina HumanHap550 quad genotyping platforms. The genotypes were imputed to the 1000 Genomes Phase 3 reference panel[54]. The height of the biological parents was measured during clinical visits. Children's adult height was measured at age 24. Measurement of height was performed using a Harpenden stadiometer (Holtain Ltd). Among the genotyped children of European ancestries who had measured adult standing height, 941 had both maternal and paternal height measures of biological parents, 1305 only had maternal height measures of biological mothers, and 151 only had paternal height measures of biological fathers (Supplementary Table 1). We generated height z-scores by separately standardizing the children's, mothers' and fathers' measured height to have zero mean and unit variance, where the standardization for the population of children was stratified by sex.

From studies that did not involve the ALSPAC cohort, we obtained an empirical estimate for $\alpha$ based on the association between a polygenic risk score for height z-score and the measured height z-score[7]. We further obtained an empirical estimate for $\beta$ based on the association between the mid-parental height z-score and the measured height z-score[32] based on Eq. (11). Subsequently, following Section Continuous trait prediction incorporating parental trait measures we derived predicted height z-score for each child by combining their calculated polygenic risk score with one or both parental height z-scores (Eq. (13) for children

with mid-parental height measures and Eq. (14) for children with only maternal or paternal height measures).

We evaluated the proportion of variance explained and the RMSE (after transforming the predictor of z-scores back to the scale of absolute height measures) of the joint predictors and compared these metrics of predictive performance with those based on the polygenic risk score, the parental height z-score alone.

Furthermore, a gold-standard predictor was obtained by fitting multivariate linear regression including both the polygenic risk score and parental height z-score as predictors on the test samples. This represents the upper bound of predictive performance achievable by any linear predictors.

**Predicting risk of complex diseases in the UK Biobank**. From 2006 to 2010, the UK Biobank study recruited approximately 500,000 participants who were aged between 40-69 years, at multiple recruitment centres in the United Kingdom[30]. Ethics approval for the UK Biobank study was obtained from the North West Centre for Research Ethics Committee (11/NW/0382). The UK Biobank ethics statement is available at https://www.ukbiobank.ac.uk/learn-more-about-uk-biobank/about-us/ethics. All UK Biobank participants provided informed consent at recruitment.

Upon recruitment, demographic and anthropometric information were collected. Genotyping of more than 480,000 participants were conducted using the Applied Biosystems™ UK BiLEVE Axiom™ Array or UK Biobank Axiom™ Array. The genotypes were imputed to the Haplotype Reference Consortium reference panel[55].

Participants who had any of the following complex diseases were identified based on inpatient International Classification of Diseases (ICD-10) diagnosis codes, Office of Population Censuses and Surveys (OPCS-4) procedure codes, or self-reported medical history during an interview with a trained nurse (Supplementary Table 2). ICD-10 codes for cancer diagnoses were retrieved by the UK Biobank through the national cancer registries. These disease outcomes included both prevalent cases identified upon initial recruitment and incident cases identified in more recent follow-up data collection.

Specifically, breast cancer included ICD-10 code C50 (malignant neoplasm of breast), specific to women; prostate cancer included ICD-10 code C61 (malignant neoplasm of prostate), specific to men; colorectal cancer included ICD-10 codes C18 (malignant neoplasm of colon), C19 (malignant neoplasm of rectal sigmoid junction), or C20 (malignant neoplasm of rectum); lung cancer included ICD-10 code C34 (malignant neoplasm of bronchus and lung); myocardial infarction included ICD-10 code I21 (acute myocardial infarction); ischemic heart disease included ICD-10 codes I20 (angina pectoris), I21 (acute myocardial infarction), I22 (subsequent myocardial infarction), I23 (complications following acute myocardial infarction), I24 (other acute ischemic heart diseases), or I25 (chronic ischemic heart disease), or OPCS-4 codes for coronary artery bypass grafting or coronary angioplasty with or without stenting; stroke included ICD-10 codes I60 (subarachnoid haemorrhage), I61 (intracerebral haemorrhage), I62 (other nontraumatic intracranial haemorrhage), I63 (cerebral infarction), or I64 (stroke, not specified as haemorrhage or infarction); type 2 diabetes included ICD-10 codes E11 (non-insulin-dependent diabetes mellitus), E13 (other specified diabetes mellitus), or E14 (unspecified diabetes mellitus), self-reported physician-made diagnosis, or self-reported use of anti-diabetic medications, excluding ICD-10 code E10 (insulin-dependent/type 1 diabetes mellitus); Alzheimer's disease included ICD-10 code G30 (Alzheimer's disease); Parkinson's disease included ICD-10 code G20 (Parkinson's disease); and COPD included ICD-10 code J44 (chronic obstructive pulmonary disease), self-reported physician-made diagnosis, or self-reported use of medications for COPD.

Upon recruitment, a questionnaire inquired whether a participant had a parental history of breast cancer, prostate cancer, bowel cancer, lung cancer, heart disease, stroke, high blood pressure, diabetes, Alzheimer's disease or dementia, Parkinson's disease, or chronic bronchitis or emphysema. Further classifications of these diseases were not available, e.g. heart disease may include various types of diseases affecting the cardiovascular system, and diabetes may include both type 1 and type 2 diabetes. Participants who responded "do not know" or "prefer not to answer" were considered missing data. No participant reported paternal history of breast cancer or maternal history of prostate cancer. We matched the participants' diseases with these parental records of diseases that had the same or a similar clinical definition.

Notably, while the disease history of siblings was also available in the UK Biobank, it lacked information on whether the sibling was a full- or half-sibling, and how many siblings were affected by the disease. Because these details were essential for correctly specifying the model, we refrained from incorporating sibling disease history in this study.

We retrieved polygenic risk scores for the above complex diseases from the PGS Catalog[33]. These polygenic risk scores were developed using different computational approaches based on source populations that did not overlap or at most slightly overlapped with the UK Biobank (Supplementary Table 3). We used the same well-powered polygenic risk score for predicting myocardial infarction and ischemic heart disease, as myocardial infarction is a complication of ischemic heart disease.

We first used a training dataset, comprising randomly selected 10% of the UK Biobank participants for deriving these parameters (Supplementary Table 2 and Supplementary Fig. 1). Specifically, for each disease, we fitted logistic regression models separately for the corresponding polygenic risk score and parental disease history (maternal disease history and paternal disease history as two independent variables), while including covariate effects of age, sex (except for breast cancer and prostate cancer), recruitment centre, genotyping array, and the first 10 genetic principal components. These two logistic regression models led to empirical estimates of $\hat{\alpha}$, $\hat{\beta}$, the baseline odds of disease $\hat{\mu}_0$, as well as the covariate effects $\hat{\gamma}$ (Section Modelling latent genetic components for binary diseases; Supplementary Table 4).

Next, we leveraged these parameters to obtain predictors of disease risk (at the liability scale; Section Binary disease risk prediction incorporating parental disease history) for the rest 90% of the UK Biobank participants (Supplementary Table 2 and Supplementary Fig. 1). We evaluated the discriminative power of these joint predictors in identifying individuals at an elevated level of disease risk by AUROC and AUPRC. We tested whether the joint predictors could more accurately quantify the genetic risk than the polygenic risk score alone by comparing their AUROC by DeLong's test[56]. We evaluated whether the risk stratification of the population could be improved by the joint predictors compared to the polygenic risk score by calculating NRI and IDI.

Last, we compared our joint predictors with a gold-standard numerical solution by jointly modelling the polygenic risk score and the parental disease history in a multivariate logistic regression model based on the training dataset (Supplementary Fig. 1), and evaluating its discriminative power on the test dataset. To assess the robustness of this data-driven approach when individual-level data were insufficient, multivariate logistic regression models were fitted on randomly sampled subsets of the training dataset of smaller sample sizes (10%, 20%, and 50% of the original training dataset, each having 100 replicates). The predictive performance of these models was also evaluated on the test dataset.

We attempted to identify sources of the under-captured genetic component for Alzheimer's disease and COPD, for which the joint predictors demonstrated the most improvements in risk stratification.

For Alzheimer's disease, we determined the *APOE* genotype for each individual in the test dataset based on genotyping data of two SNPs: rs429358 and rs7412. We tested whether the polygenic risk score and the joint predictor were associated with carrying at least one e4 allele or carrying two e4 alleles, respectively. For COPD, we tested whether the polygenic risk score and the joint predictor were associated with self-reported smoking status (ever-smokers vs. never-smokers), respectively. All association tests were based on logistic regression, adjusted for the effects of age, sex, genotyping array, recruitment centre, and the first 10 genetic principal components.

**Reporting summary**. Further information on research design is available in the Nature Research Reporting Summary linked to this article.

## Data availability

Individual genotype and pheontype data from the UK Biobank (https://www.ukbiobank.ac.uk/) and the ALSPAC (http://www.bristol.ac.uk/alspac/) are available through successful applications to the research committees. The ALSPAC website contains details of all the data that is available through a fully searchable data dictionary and variable search tool (http://www.bristol.ac.uk/alspac/researchers/our-data/). Source data underlying Figs. 2–3 are presented in Supplementary Data 1-2. All other data are available from the corresponding author on reasonable request.

## Code availability

All computational scripts for analyses in this study are available from the corresponding author on reasonable request. A computational toolkit[57] implementing the latent factor model developed in this study is available at https://github.com/tianyuan-lu/PRS-FH-Prediction.

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

## Acknowledgements

This research has been conducted using the UK Biobank resource under Application Number 27449 and 60755, and the ALSPAC resource under Application Number B3359. ALSPAC children were genotyped using the Illumina HumanHap550 quad chip genotyping platforms by Sample Logistics and Genotyping Facilities at Wellcome Sanger Institute and LabCorp (Laboratory Corporation of America), using support from 23andMe. We thank all the families who took part in the ALSPAC study, the midwives for their help in recruiting them, and the whole ALSPAC team, which includes interviewers, computer and laboratory technicians, clerical workers, research scientists, volunteers, managers, receptionists, and nurses. This study was enabled in part by support provided by Calcul Québec and Compute Canada. T.L. thanks Wenmin Zhang for helpful discussion about the latent factor models and prediction schemes. C.M.T.G. is supported by a Canadian Institutes of Health Research grant (CIHR; PJT-148620). The J.B.R. research group is supported by the Canadian Institutes of Health Research (365825; 409511), the Lady Davis Institute of the Jewish General Hospital, the Canadian Foundation for Innovation, the NIH Foundation, Cancer Research UK, Genome Québec, the Public Health Agency of Canada, and the Fonds de Recherche Québec Santé (FRQS). J.B.R. is supported by a FRQS Clinical Research Scholarship Merite. T.L. has been supported by a Vanier Canada Graduate Scholarship, an FRQS Doctoral Training

Fellowship, and a McGill University Faculty of Medicine Scholarship. The UK Medical Research Council and Wellcome (217065/Z/19/Z) and the University of Bristol provide core support for ALSPAC. A comprehensive list of grants funding is available on the ALSPAC website (http://www.bristol.ac.uk/alspac/external/documents/grant-acknowledgements.pdf). This research was specifically funded by Wellcome Trust and the Medical Research Council (076467/Z/05/Z).

## Author contributions

T.L. conceived and designed this study. T.L. and C.M.T.G. developed methodology. T.L. created software package. V.F. and T.L. acquired and managed data. T.L. performed analyses and visualized results. T.L. interpreted results with C.M.T.G., J.B.R., and V.F. C.M.T.G. supervised this study. T.L. drafted the manuscript. All authors read and revised the manuscript.

## Competing interests

The authors declare the following competing interests: J.B.R. is the founder of 5 Prime Sciences, and has served as a consultant to GlaxoSmithKline and Deerfield Capital for their genetics programs. The remaining authors declare no competing interests.
