## [Peer Review File · Communications Biology]

Reviewers' comments:

Reviewer #1 (Remarks to the Author):

Lu et al present a method for improving prediction of complex traits by incorporating genetic information (from polygenic risk scores; PRS) with family history (specifically, the trait values for an individual's mother or father). Overall, I am impressed by this paper. The results for height (substantially better prediction compared to just PRS or just family history) seem strong, while the results for the diseases (marginally better prediction) are non-trivial.

To answer the questions in the review form:

The work is convincing and the statistics seem valid - particularly, the mathematics behind the new tool seem novel and assured.

The new tool could prove useful (with the caveats below).

Major comments

The main shortcoming is that the paper does not compare their new approach to any existing approaches (aside from the very naive approach of using mid-parent height). Are there really no existing methods for incorporating family history? From reading the paper, I wondered whether this paper (<https://www.nature.com/articles/s41467-020-16829-x>) would this be an alternative. Alternatively, while this paper (<https://pubmed.ncbi.nlm.nih.gov/32313248/>) considers association testing (and not directly prediction), it seems to contain similar ideas. Further, I can imagine there are methods in animal breeding that are able to incorporate partial pedigree information (which seems a closely related problem).

While the improvement for height is very impressive, the improvement for the 11 diseases is consistent (and across all diseases, likely significant), but not spectacular. I appreciate the authors have thought about this, for example, by constructing and comparing to a gold standard, but I think this limitation should be more clear to the reader. For example, the abstract says the paper considers 12 traits, but reports only the performance for height, giving the impression (at least to me) that this accuracy is representative for all 12 traits, which is a bit misleading. Further, I think the discussion should mention the difference in performance between height and the diseases.

Continuing the previous point, perhaps you could, for example, consider additional quantitative traits (eg BMI) to determine if the difference in improvement is due to differences between height and the diseases, or differences between predicting quantitative traits and binary traits.

Minor comments

It seemed a drawback that you required a separate dataset (ie 10% of UKBB) in order to first estimate model parameters (alpha and beta), because this stops people directly using your method to construct predictions for a new sample of individuals. However, I could not see why this was the case. I believe you should be able to estimate alpha (accuracy of PRS) fairly easily, using for example the GWAS used to construct the PRS (e.g., if I wanted to do this, I would use my pseudo-summary statistics, but there are probably other ways). To estimate beta, you just need family data? or estimates of total heritability? How accurate do you need your estimates of alpha and beta (in order to maintain an improvement in PRS accuracy).

On Line 218, you describe alpha and beta as invariant for height. However, for the binary traits, you suggested these were population specific (line 397). I can understand both (because height is an "easy" trait to measure, whereas the diseases are not, and different definitions can lead to different heritabilities). However, I thought this could be more clear (i.e., if I understand correctly, make it

clear in Line 218 that your assumption is specific to height, or to quantitative traits, while in Line 397, it is specific to diseases, etc).

In your methods, you say that with P and G standardized, $\alpha^2 + \beta^2$ equals trait heritability. However, for height, your estimates of alpha and beta (in supp table 4) correspond to a heritability of just over 1. This was a bit concerning, please can you clarify. Likewise, the estimates for binary traits were often over one (but in that case, it might reflect difference between observed and liability scale??)

Consider giving the tool a name so that people can more easily refer to your method (I note the github page calls it PRS-FH)

Line 394. "not strongly overlap" suggests (to me) maybe 10% overlap. However, in practice, you probably have only a tiny overlap (especially as many of the GWAS are not UK-based). In other papers, I would see this written as "no known overlap" (maybe adding in parentheses, that some chance overlap is possible).

I really like the maths behind your version for binary traits. However, could you simply compare PRS using the linear approximation (ie just treat binary traits as if they are linear) to see whether the clever logistic model gives much value?

Finally, am I correct in thinking you require data for multiple individuals in order to fully utilise your method? As in, if you only had one individual (e.g., you were a doctor, with an individual's genotypes and family history), your tool would probably perform poorly (no better than just using the PRS).

Signed Doug Speed

ps, sorry I have been a bit slow with this review (23 days).

Reviewer #2 (Remarks to the Author):

This is an interesting paper, showing that family information can increase the prediction accuracy in the context of polygenic risk score prediction. The authors developed a latent factor model that integrates polygenic effects and family history for a genetic prediction of complex traits and diseases. The authors argue that their finding supports incorporation of family history into polygenic risk score-based genetic risk prediction models.

I think this paper will potentially have a significant impact in the field. I have a few suggestions to improve the current version of the manuscript.

It is confusing which summary stats should be used. I think the summary stats should include joint estimates based on polygenic and family history components. Or, can the method use GWAS summary stats alone?

It is also confusing and not clear when individual-level data are required. Didn't authors use Individual genotypes and phenotypes for the target dataset? Should the target samples have family history information as well? Did the authors compare the prediction accuracy with and without family information available in target samples?

These should be made clearer in the text (e.g. section 2.1).

I would suggest the authors should include relevant literature that is currently missing. There are a number of studies demonstrating family information can increase the prediction accuracy, e.g.

Efficient polygenic risk scores for biobank scale data by exploiting phenotypes from inferred relatives. Nature Communications volume 11, Article number: 3074 (2020)

To get GWAS summary stats integrating family information, there are several relevant studies in animal genetics, e.g.

Frequentist p-values for large-scale-single step genome-wide association, with an application to birth weight in American Angus cattle. Genetics Selection Evolution volume 51, Article number: 28 (2019)

Although this was not applied to human population, the models may be related, for which the authors should discuss.

Reviewer #3 (Remarks to the Author):

Lu and Colleagues present a new and interesting framework to calculate an improved polygenic risk score (PRS) that takes into account common genetic variants effects as well as unobserved intra-family genetic/exposure variability.

General comments:

Overall the research presented is innovative and has the potential to improve the current state of PRS research and application. The manuscript is well presented, concise and clear.

The presented methodology makes a strong assumption that is largely untestable. However, this assumption enables for simple estimation of the main parameters and final formula as beautifully presented in the manuscript methods section.

The results are convincing and nicely presented. An R-software toolkit is provided by the authors to aid implementation for future users.

Discussions are well presented.

Specific comments:

The authors specified that the latent parental contribution included alongside the individual PRS is also genetic. However, the authors should consider the contribution of parental phenotypic information captures both unseen genetic as well as intra-familial environmental/cultural contribution. As a dummy example, meat consumption might be causally associated with height, parents who culturally consume a large amount of meat are likely to be tall and likely to feed their offspring a large amount of meat. Meat consumption in this case is environmental/cultural but will be nonetheless captured by the latent variable presented in this work.

Additionally, the authors should mention in the discussion that although the data required by this tool (parental history) might be collected in clinics for risk prediction estimation purposes, this is not the case in many genetic studies where unrelated people are often preferred.

Throughout the manuscript, this reviewer wondered whether it would be possible (as future

developments of this tool) to generalise this approach and include phenotypic information related to all distant relatives (as estimated via genetic relationship scores) when parental information is not available for all subjects (many traits of UK Biobank do not have any information about family history).

Discussion of further development or improvements (see for example above comment) for this methodology is also missing.

Lastly, this reviewer recommends this manuscript for publication and only has some minor suggestions:

Please mention in the introduction that an R-toolkit is made available. Additionally, specify in the results or discussion what variables the user will require to use the tool and how to obtain them.

In the github repository, a more detailed step by step guide on how to use the tool and how to estimate the needed parameters using a toy dataset would be very useful

Please mention whether the tool captures only unseen genetic contribution or also environmental (see above comment).

The assumption that this tool makes is largely untestable, however, simulation data could help understand how robust the current implementation is to deviation from such assumption. Something simple like simulated normally distributed trait such as height would suffice.

Please include potential future development/improvement in the discussion section

Finally, this reviewer would like to congratulate the author for the wonderful work presented here.

Best Regards,

Roberto Bonelli, PhD

Research Officer

Walter & Eliza Hall Institute of Medical Research

Reviewers' comments:

Reviewer #1 (Remarks to the Author):

Lu et al present a method for improving prediction of complex traits by incorporating genetic information (from polygenic risk scores; PRS) with family history (specifically, the trait values for an individual's mother or father). Overall, I am impressed by this paper. The results for height (substantially better prediction compared to just PRS or just family history) seem strong, while the results for the diseases (marginally better prediction) are non-trivial.

To answer the questions in the review form:

The work is convincing and the statistics seem valid - particularly, the mathematics behind the new tool seem novel and assured. The new tool could prove useful (with the caveats below).

-We thank the reviewer for these comments.

Major comments

The main shortcoming is that the paper does not compare their new approach to any existing approaches (aside from the very naive approach of using mid-parent height). Are there really no existing methods for incorporating family history? From reading the paper, I wondered whether this paper (<https://www.nature.com/articles/s41467-020-16829-x>) would this be an alternative. Alternatively, while this paper (<https://pubmed.ncbi.nlm.nih.gov/32313248/>) considers association testing (and not directly prediction), it seems to contain similar ideas. Further, I can imagine there are methods in animal breeding that are able to incorporate partial pedigree information (which seems a closely related problem).

-We appreciate these suggestions. We have added a "Related methods" section at the beginning of Methods discussing these and a few other previous studies:

"Combining a genetic risk score with measures of traits or disease status of relatives is not unprecedented. The most straightforward approach is to fit a multivariate regression model including both the genetic risk score and family history variables as predictors (Lu et al.; You et al.). Alternatively, extensions of the best linear unbiased prediction (BLUP) method have been proposed by appending an empirical relatedness matrix based on pedigree information to a genotype-based relatedness matrix for modelling random effects in a mixed model setting (Truong et al.; Tucker et al.). Using a similar framework, family history may also improve the power of genetic association tests (Hujoel et al.). However, these methods require access to a training dataset that simultaneously contains phenotypes, genotypes, and family history information. For modelling disease outcomes, a large number of cases is needed to ensure statistical power. This is often unlikely due to confidentiality restrictions or logistical constraints. In addition, the variance-covariance structure of genetic components specified in our method is also similar to that implemented in a few methods using family disease status to modify genetic

risk estimates in known risk loci (Lencz et al.; So et al.; Ruderfer et al.). Nonetheless, these methods require a pre-specified estimate of disease heritability, which may be prone to error if no reference for the targeted population is available or if unmeasured covariate effects are not properly accounted for.”

Throughout the manuscript, we have removed the claims of novelty as per journal regulations.

We would like to highlight that the major drawback of these methods, as we mentioned in Introduction, is the requirement of individual-level training data to train new polygenic risk scores or covariate-adjusted polygenic risk scores from scratch.

“Nevertheless, creating accurate joint predictors may be challenging, since it requires modelling individual-level training data on phenotypes, genotypes, and family history information, as has been explored previously (Lu et al.; You et al.; Truong et al.; Tucker et al.). This may not lead to effective prediction models if datasets containing all the required information are too small, particularly for diseases with a low prevalence in the population.”

To illustrate this, we have implemented the HBLUP method as the reviewer suggested, using the same 10% UK Biobank as the training dataset. In Additional Figure 1 below, we compare the performance of GBLUP (Genomic BLUP, the foundation of HBLUP) scores, HBLUP scores, polygenic risk scores we used and the FHPRS created by our method based on the UK Biobank test dataset. We found that both the GBLUP and HBLUP methods were worse than the polygenic risk scores we used even without adding family history information.

This result is expected, because the polygenic risk scores we retrieved from the PGS Catalog were developed based on large-scale meta-analyses of case-control GWASs, which included much more cases than in the UK Biobank. To perform a fair comparison, both the GBLUP and HBLUP scores should have been constructed based on genotype and phenotype data from all cohorts participating in each GWAS. But the fact is, most researchers do not have access to these individual-level data, rendering it impossible for them (us included) to maximize the genetic predictive power by building polygenic risk scores from scratch. We posit that this is one of the major advantages of our method, since it can create multivariate predictive models in the absence of any individual-level training data.

Additional Figure 1. Performance comparison based on the UK Biobank test dataset. FHPRS: family history-assisted polygenic risk score; GBLUP: genomic best linear unbiased predictor; HBLUP: H-matrix best linear unbiased predictor; PRS: polygenic risk score (from PGS Catalog).

While the improvement for height is very impressive, the improvement for the 11 diseases is consistent (and across all diseases, likely significant), but not spectacular. I appreciate the authors have thought about this, for example, by constructing and comparing to a gold standard, but I think this limitation should be more clear to the reader. For example, the abstract says the paper considers 12 traits, but reports only the performance for height, giving the impression (at least to me) that this accuracy is representative for all 12 traits, which is a bit misleading. Further, I think the discussion should mention the difference in performance between height and the diseases.

-We thank the reviewer for pointing this out. First of all, we have revised the abstract to avoid the misleading information:

“Family history of complex traits may reflect transmitted rare pathogenic variants, intra-familial shared exposures to environmental and lifestyle factors, as well as a common genetic predisposition. We developed a latent factor model to quantify trait heritability in excess of that captured by a common variant-based polygenic risk score, but inferable from family history. For 941 children in the Avon Longitudinal Study of Parents and Children cohort, a joint predictor combining a polygenic risk score for height and mid-parental height was able to explain ~55% of the total variance in sex-adjusted adult height z-scores, close to the estimated heritability. **Marginal yet consistent risk prediction improvements were also achieved among ~400,000 European ancestry participants for 11 complex diseases in the UK Biobank.** Our work showcases an innovative paradigm for risk calculation, and supports incorporation of family history into polygenic risk score-based genetic risk prediction models.”

On the other hand, we posit that the performance metrics are not comparable between the continuous and binary outcomes. It has been recognized that metrics of discriminative power, such as the AUC, are not sensitive to addition of new predictors, even if the additional predictors are useful and are independent of the baseline predictive model. Therefore, we have investigated the joint predictors’ discriminative power from several different perspectives – AUROC, AUPRC, NRI, IDI, as well as the distribution of predicted risks.

Following the reviewer’s suggestion, we have highlighted the differences between height prediction and disease risk prediction in Discussion:

“The combination of parental height measures and a polygenic risk score for height brought substantial improvements in adult height prediction. In fact, the proportion of variance explained in adult height z-scores by our joint predictor in the ALSPAC cohort was similar to the estimated total SNP heritability of height z-scores (Yengo et al.; Yang et al.; Yang et al.; Yang et al.). On the other hand, for 11 complex diseases, the joint predictors consistently demonstrated stronger discriminative power in identifying individuals at an elevated level of risk, although the improvements appeared to be marginal. This may reflect limited sensitivity of relevant metrics to addition of new predictors (Baker et al.), as we observed in simulation studies (Supplementary Notes). Nonetheless, by jointly examining all metrics (AUROC, AUPRC, NRI, and IDI) as well as the distribution of predicted risks, we posit that the resulting risk re-stratification could still benefit up to thousands of individuals at the biobank-scale.”

To further illustrate such differences, we conducted simulation studies to investigate the joint predictors’ performance compared to predictions using polygenic risk scores alone. Details and full results of these simulations are provided in Supplementary Notes.

We compared the performance of joint predictors built using our model versus directly using polygenic risk scores in Additional Figure 2 below. For continuous traits, we calculated

$$\Delta_{metric} = R_{FHPRS}^2 - R_{PRS}^2$$

For binary outcomes, we calculated

$$\Delta_{metric} = AUROC_{FHPRS} - AUROC_{PRS}$$

We found that under the same model setting (proportions of trait variance attributable to the genetic components), the increase in R^2 for predicting a continuous trait would always be larger than the increase in $AUROC$ for predicting a binary outcome.

Figure S4. Comparison of performance between a joint predictor and polygenic risk score. Model parameters for each simulation setting are indicated. Δ_{metric} was defined as difference in proportion of variance explained for the simulated continuous trait and difference in area under the receiver operating characteristic curve for the simulated binary outcome based on the test dataset. A positive Δ_{metric} indicated that the joint predictor had improved predictive performance compared to the polygenic risk score. Each simulation setting was repeated 100 times to obtain the distribution of Δ_{metric} .

Continuing the previous point, perhaps you could, for example, consider additional quantitative traits (eg BMI) to determine if the difference in improvement is due to differences between height and the diseases, or differences between predicting quantitative traits and binary traits.

-We appreciate the reviewer's suggestion. However, for quantitative traits, the UK Biobank does not have sufficient data for parental phenotypic values. We considered the ALSPAC cohort yet our data access was associated with an application (link below) that had specific clinical objectives upon childhood and adolescence height monitoring and management

<https://proposals.epi.bristol.ac.uk/?q=node/129958>.

Unfortunately, we do not have access to weight information nor can we request additional data in the same application for the purpose of method evaluation. We posit that the simulation studies presented above and in Supplementary Notes are better suited for illustrating that the differences between predicting height and the diseases – (1) a continuous trait is indeed easier to predict and (2) metrics for evaluating binary outcome predictions are more difficult to improve.

Minor comments

It seemed a drawback that you required a separate dataset (ie 10% of UKBB) in order to first estimate model parameters (alpha and beta), because this stops people directly using your method to construct predictions for a new sample of individuals. However, I could not see why this was the case. I believe you should be able to estimate alpha (accuracy of PRS) fairly easily, using for example the GWAS used to construct the PRS (e.g., if I wanted to do this, I would use my pseudo-summary statistics, but there are probably other ways). To estimate beta, you just need family data? or estimates of total heritability? How accurate do you need your estimates of alpha and beta (in order to maintain an improvement in prs accuracy).

-This is an important question. Using real data, we do not know whether our model parameter estimates are accurate and to what extent errors in parameter estimation can lead to attenuated model performance.

We have conducted simulation studies to better characterize the robustness of our method. Details and full results of these simulations are provided in Supplementary Notes. Briefly, we supplied inaccurate parameters to our model

$$\begin{aligned}\tilde{\alpha} &= \delta_{\alpha}\alpha \\ \tilde{\beta} &= \delta_{\beta}\beta\end{aligned}$$

where α and β are true parameters used in simulation and the error factors δ_{α} and δ_{β} take value in $\{0.5,0.8,0.9,1.1,1.2,1.5\}$. In Additional Figure 3 below, we illustrate that the model performance remained largely unchanged with up to 20% under- or over-estimation of α and β .

Additional Figure 3. Impact of error in parameter estimation with a strong under-modelled genetic component ($\beta = 0.5$). The difference in proportion of variance explained for the simulated continuous trait based on the test dataset was calculated under different degrees of parameter estimation error, specified by error factors δ_α and δ_β (Supplementary Notes). Each simulation setting was repeated 100 times. FHPRS: family history-assisted polygenic risk score; PRS: polygenic risk score.

To obtain β , we need the magnitude of association between parents' trait measures or disease history and the target trait or disease amongst children. However, we decided to use 10% of the UK Biobank as our reference dataset, instead of another cohort, because we need to ensure the reference and test datasets were similar in genetic ancestry, demographic characteristics, disease prevalence, etc. This does not preclude the utility of our method in other cohort studies as long as these prerequisites upon reference dataset for parameter estimation are satisfied. We discuss our considerations in response to the next question below.

On Line 218, you describe alpha and beta as invariant for height. However, for the binary traits, you suggested these were population specific (line 397). I can understand both (because height is an "easy" trait to measure, whereas the diseases are not, and different definitions can lead to different heritabilities). However, I thought this could be more clear (i.e., if I understand correctly, make it clear in Line 218 that your assumption is specific to height, or to quantitative traits, while in Lind 397, it is specific to diseases, etc).

-We apologize for this confusion. In fact, for both continuous and binary traits, we assume that the magnitudes of genetic effects are constant among the parents and among the children. We have clarified the first instance:

“We further assume that the effects of the modelled polygenic component and the under-captured genetic component, α and β , are invariant in the populations of parents and children from the same study.”

However, we would expect parameters to vary in different study populations, e.g. between the UK Biobank and another cohort study that has a different genetic ancestry composition, age distribution, disease prevalence, or socioeconomic status etc. This was the main reason why our analyses used 10% of the UK Biobank as reference population to obtain the model parameters, since it would ensure the reference population and the test population were largely homogeneous. We have removed the original description and have added in Discussion:

“Last, our findings should be considered specific to the study populations. Participants in our study cohorts are predominantly of European ancestries, yet it has been widely recognized that polygenic risk scores can have largely attenuated predictive performance when applied to populations of different genetic ancestries. Hence, we recommend extensive validations of our results in diverse populations as well as development of ancestry-specific polygenic risk scores. Adding to this restriction, we also note that participants in the UK Biobank have been shown to be slightly healthier, less obese, and less likely to smoke and drink alcohol than the general population in the United Kingdom. In particular, the average age of the study cohort was substantially younger than the average age of onset for Alzheimer's disease and Parkinson's disease, thus the corresponding disease prevalence was low. Therefore, if a joint predictor were constructed for other populations, we strongly recommend estimating model parameters based on reference cohort studies with similar demographic characteristics and prevalence of disease.”

In your methods, you say that with P and G standardized, $\alpha^2 + \beta^2$ equals trait heritability. However, for height, your estimates of alpha and beta (in supp table 4) correspond to a heritability of just over 1. This was a bit concerning, please can you clarify. Likewise, the estimates for binary traits were often over one (but in that case, it might reflect difference between observed and liability scale??)

-We sincerely appreciate the reviewer’s careful examination. This was a typo – α is estimated by taking the square root of the proportion of trait variance explained by the polygenic risk score, which in this case is $\sqrt{0.367} = 0.606$, instead of 0.656. We attach the corrected table below.

Continuous trait							
	Proportion of trait variance captured by polygenic risk score	Proportion of variance explained by mid-parental height	$\hat{\alpha}$	$\hat{\beta}$			Proportion of trait variance not captured by polygenic risk score

Height z-score	0.367	0.449	0.606	0.762		0.581
----------------	-------	-------	-------	-------	--	-------

For continuous traits, $\alpha^2 + \beta^2$ (0.948 for height z-score above) should not exceed 1, otherwise violations of model assumptions would be suspected. For binary traits, as the reviewer mentioned, $\alpha^2 + \beta^2$ do not directly represent heritability and are unbounded, because heritability estimation in a liability model is different, as shown in Equations 17 and 18.

We have clarified in Methods:

“Consequently, heritability captured by the polygenic risk score can be estimated as τ_α^2 , while the under-captured heritability can be estimated as τ_β^2 . Notably, unlike the model for continuous traits where α and β are automatically bounded ($0 \leq \alpha^2 + \beta^2 \leq 1$), here, the effects of the genetic components P and G are unconstrained.”

Consider giving the tool a name to that people can more easily refer to your method (I note the github page calls it PRS-FH)

-We thank the reviewer for this suggestion. We have now mentioned in Introduction:

“An R toolkit implementing the method developed in this work, called FHPRS (Family History-assisted Polygenic Risk Score), is openly available at <https://github.com/tianyuan-lu/PRS-FH-Prediction>.”

Line 394. "not strongly overlap" suggests (to me) maybe 10% overlap. However, in practice, you probably have only a tiny overlap (especially as many of the gwas are not UK-based). In other papers, I would see this written as "no known overlap" (maybe adding in parentheses, that some chance overlap is possible).

-We appreciate this suggestion. We have rephrased this sentence:

“These polygenic risk scores were developed using different computational approaches based on source populations that did not overlap or at most slightly overlapped with the UK Biobank.”

I really like the maths behind your version for binary traits. However, could you simply compare PRS using the linear approximation (ie just treat binary traits as if they are linear) to see whether the clever logistic model gives much value?

-We have compared the continuous approximation with the prediction derived from logistic model below, in Additional Figure 4. As expected, the continuous approximation generated predictors whose performance lies between of the logistic model-derived predictors (FHPRS) and the raw polygenic risk scores, except for stroke, where the continuous approximation and FHPRS have the same AUROC. We therefore believe that the logistic model does perform better in modelling binary outcomes.

Additional Figure 4. Comparison of model performance in the UK Biobank test dataset.

Finally, am I correct in thinking you require data for multiple individuals in order to fully utilise your method? As in, if you only had one individual (e.g., you were a doctor, with an individual's genotypes and family history), your tool would probably perform poorly (no better than just using the PRS).

-Indeed, our method does require reference data for obtaining model parameters. If such data had been available, the tool would be ready to be used in a clinical setting, even if there is only one new patient. However, if no reference is available or if the reference datasets do not match the characteristics of a patient to be tested, the method would have limited utility, as we discussed above.

Signed Doug Speed

ps, sorry I have been a bit slow with this review (23 days).

-Again, we thank Dr. Speed for these helpful comments and suggestions.

Reviewer #2 (Remarks to the Author):

This is an interesting paper, showing that family information can increase the prediction accuracy in the context of polygenic risk score prediction. The authors developed a latent factor model that integrates polygenic effects and family history for a genetic prediction of complex traits and diseases. The authors argue that their finding supports incorporation of family history into polygenic risk score-based genetic risk prediction models.

I think this paper will potentially have a significant impact in the field. I have a few suggestions to improve the current version of the manuscript.

-We thank the reviewer for these comments.

It is confusing which summary stats should be used. I think the summary stats should include joint estimates based on polygenic and family history components. Or, can the method use GWAS summary stats alone? It is also confusing and not clear when individual-level data are required. Didn't authors use Individual genotypes and phenotypes for the target dataset? Should the target samples have family history information as well? Did the authors compare the prediction accuracy with and without family information available in target samples? These should be made clearer in the text (e.g. section 2.1).

-We thank the reviewer for pointing this out and apologize for this confusion. By summary statistics, we meant the magnitudes of associations between the target traits and polygenic risk scores or family history, instead of GWAS summary statistics. We do not need individual-level data for training the predictive model, yet we do need individual-level data (both genotype and family information) for generating predictions for a test dataset. We did compare our method with the predictors built with and without family information (Table 1).

We have clarified in Introduction:

“Therefore, in this work, we demonstrated both by theory and with examples, the improved predictions associated with a new scheme for combining polygenic risk scores and parental disease histories. We approached this goal by inquiring what proportion of trait variance is captured by parental information and not by existing polygenic risk scores. Then, in comparison to using polygenic risk scores alone or predictors based only on parental trait measures or parental disease history, we evaluated the performance of these joint predictors in predicting adult height amongst 2,397 European ancestry children in the Avon Longitudinal Study of Parents and Children (ALSPAC) cohort, as well as in predicting risk for 11 complex diseases amongst ~400,000 European ancestry participants in the UK Biobank.”

And in Results:

“Based on this model, a multivariate predictive model for a continuous trait or the genetic liability for a disease can be created by combining a polygenic risk score and family history, requiring only (1) the magnitude of association between a polygenic risk score and the target trait or disease, and (2) the magnitude of association between parents’ trait measures or disease history and the target trait or disease amongst children. Importantly, these estimates can be obtained from separate well-powered reference cohort studies, without the need to access individual-level information for training the predictive model. Subsequently, predictions for individuals in a test population or patients in clinics can be obtained based on their genotypes and parental trait measures or family disease history.”

I would suggest the authors should include relevant literature that is currently missing. There are a number of studies demonstrating family information can increase the prediction accuracy, e.g.

Efficient polygenic risk scores for biobank scale data by exploiting phenotypes from inferred relatives. *Nature Communications* volume 11, Article number: 3074 (2020)

To get GWAS summary stats integrating family information, there are several relevant studies in animal genetics, e.g. Frequentist p-values for large-scale-single step genome-wide association, with an application to birth weight in American Angus cattle. *Genetics Selection Evolution* volume 51, Article number: 28 (2019)

Although this was not applied to human population, the models may be related, for which the authors should discuss.

-We appreciate the relevant literature suggested by the reviewer. We have added a “Related methods” section at the beginning of Methods discussing these and a few other previous studies:

“Combining a genetic risk score with measures of traits or disease status of relatives is not unprecedented. The most straightforward approach is to fit a multivariate regression model including both the genetic risk score and family history variables as predictors (Lu et al.; You et al.). Alternatively, extensions of the best linear unbiased prediction (BLUP) method have been proposed by appending an empirical relatedness matrix based on pedigree information to a genotype-based relatedness matrix for modelling random effects in a mixed model setting (Truong et al.; Tucker et al.). Using a similar framework, family history may also improve the power of genetic association tests (Hujoel et al.). However, these methods require access to a training dataset that simultaneously contains phenotypes, genotypes, and family history information. For modelling disease outcomes, a large number of cases is needed to ensure statistical power. This is often unlikely due to confidentiality restrictions or logistical constraints. In addition, the variance-covariance structure of genetic components specified in our method is also similar to that implemented in a few methods using family disease status to modify genetic risk estimates in known risk loci (Lencz et al.; So et al.; Ruderfer et al.). Nonetheless, these methods require a pre-specified estimate of disease heritability, which may be prone to error if

no reference for the targeted population is available or if unmeasured covariate effects are not properly accounted for.”

Throughout the manuscript, we have removed the claims of novelty as per journal regulations.

We would like to highlight that the major drawback of these methods, as we mentioned in Introduction, is the requirement of individual-level training data to train new polygenic risk scores or covariate-adjusted polygenic risk scores from scratch.

“Nevertheless, creating accurate joint predictors may be challenging, since it requires modelling individual-level training data on phenotypes, genotypes, and family history information, as has been explored previously. This may not lead to effective prediction models if datasets containing all the required information are too small, particularly for diseases with a low prevalence in the population.”

Reviewer #3 (Remarks to the Author):

Lu and Colleagues present a new and interesting framework to calculate an improved polygenic risk score (PRS) that takes into account common genetic variants effects as well as unobserved intra-family genetic/exposure variability.

General comments:

Overall the research presented is innovative and has the potential to improve the current state of PRS research and application. The manuscript is well presented, concise and clear.

The presented methodology makes a strong assumption that is largely untestable. However, this assumption enables for simple estimation of the main parameters and final formula as beautifully presented in the manuscript methods section.

The results are convincing and nicely presented. An R-software toolkit is provided by the authors to aid implementation for future users.

Discussions are well presented.

-We thank the reviewer for these comments.

Specific comments:

The authors specified that the latent parental contribution included alongside the individual PRS is also genetic. However, the authors should consider the contribution of parental phenotypic information captures both unseen genetic as well as intra-familial environmental/cultural contribution. As a dummy example, meat consumption might be causally associated with height, parents who culturally consume a large amount of meat are likely to be tall and likely to feed

their offspring a large amount of meat. Meat consumption in this case is environmental/cultural but will be nonetheless captured by the latent variable presented in this work.

-We completely agree with the reviewer. We have specified in the first section of Results:

“The under-captured genetic component could include the effects of unmeasured common variants, rare variants, gene-by-environment interactions, epistasis, intra-familial shared environmental or lifestyle factors, etc.”

And in Discussion with reference to the results of COPD analysis, where the joint predictor was associated with smoking while the polygenic risk score was not:

“Furthermore, by design, inherited risk factor exposure was also included in the under-captured genetic component. Smoking is one of the most important risk factors for COPD and is heritable. In our model, family disease history of COPD may partially capture the genetic predisposition to smoking as well as other risk factors that is not fully represented in the polygenic risk score. As a result, the corresponding joint predictor aggregating additional genetic risks demonstrated a prominent improvement over the polygenic risk score.”

Additionally, the authors should mention in the discussion that although the data required by this tool (parental history) might be collected in clinics for risk prediction estimation purposes, this is not the case in many genetic studies where unrelated people are often preferred.

-We have mentioned this limitation in Discussion:

“if a joint predictor were constructed for other populations, we strongly recommend estimating model parameters based on reference cohort studies with similar demographic characteristics and prevalence of disease. This may be particularly important for large cohort studies and population-level screening programs where family history information is less comprehensive than in clinical settings.”

Throughout the manuscript, this reviewer wondered whether it would be possible (as future developments of this tool) to generalise this approach and include phenotypic information related to all distant relatives (as estimated via genetic relationship scores) when parental information is not available for all subjects (many traits of UK Biobank do not have any information about family history).

-Our method is indeed generalizable to include distant relatives, but we posit that the utility of adding such less directly relevant information should be examined carefully in studies with more comprehensive pedigree information. We have mentioned in Methods:

“This framework is generalizable to any degree of family relationships, with modification of the variance-covariance matrices, possibly using empirical estimates of genetic relatedness.”

“However, because information about more distantly related family members is more difficult to obtain, and is rarely complete in large cohort studies, and because parental information is the most relevant to risk prediction, we focus on parental trait measures or disease history in this work.”

Further discussions are included in response to the next comment.

Discussion of further development or improvements (see for example above comment) for this methodology is also missing.

-We appreciate this suggestion. We have described three possible improvements in Discussion:

“These results not only support the utility of family history in enhancing predictive power, but also encourage explicit modelling of large effects such as monogenic causes or significant intra-familial shared risk factor exposures in polygenic risk prediction.”

“We anticipate that the predictive performance of the joint predictors could be further enhanced with additional knowledge of family history, such as the disease history of relatives other than the parents, using empirical genetic relatedness based on pedigree information. However, we expect that information gained from second-degree or more distant relatives would be less significant compared to first-degree relatives.”

“Furthermore, in the UK Biobank, the parental disease history largely reflected the parents' lifetime risk given that the participants (children) were aged above 40 years upon recruitment. Hence, appropriate modelling of age-dependent risks should be pursued for most complex diseases that do not have an early onset, if disease history of younger relatives were to be considered.”

Improvements upon model specification have been discussed in Supplementary Notes and described below in response to the reviewer's suggestion on evaluating model robustness via simulation.

Lastly, this reviewer recommends this manuscript for publication and only has some minor suggestions:

Please mention in the introduction that an R-toolkit is made available. Additionally, specify in the results or discussion what variables the user will require to use the tool and how to obtain them.

-We thank the reviewer for this suggestion. We have mentioned in Introduction:

“An R toolkit implementing the method developed in this work, called FHPRS (Family History-assisted Polygenic Risk Score), is openly available at <https://github.com/tianyuan-lu/PRS-FH-Prediction>.”

We have specified the required inputs in the first section of Results:

“Based on this model, a multivariate predictive model for a continuous trait or the genetic liability for a disease can be created by combining a polygenic risk score and family history, requiring only (1) the magnitude of association between a polygenic risk score and the target trait or disease, and (2) the magnitude of association between parents’ trait measures or disease history and the target trait or disease amongst children. Importantly, these estimates can be obtained from separate well-powered reference cohort studies, without the need to access individual-level information for training the predictive model. Subsequently, predictions for individuals in a test population or patients in clinics can be obtained based on their genotypes and parental trait measures or family disease history.”

More detailed information and a tutorial have been included in the github repository, as the reviewer suggested.

In the github repository, a more detailed step by step guide on how to use the tool and how to estimate the needed parameters using a toy dataset would be very useful. Please mention whether the tool captures only unseen genetic contribution or also environmental (see above comment).

-We appreciate these suggestions. We have updated the github repository. A detailed tutorial has been included in the “Data example” section with a simulated reference dataset for estimating parameters and a simulated test dataset for assessing prediction.

We highlight in bold in the “Method overview” section that the tool does potentially capture environmental factors:

“The under-captured genetic component could include the effects of unmeasured common variants, rare variants, gene-by-environment interactions, epistasis, intra-familial shared environmental or lifestyle factors, etc.”

The assumption that this tool makes is largely untestable, however, simulation data could help understand how robust the current implementation is to deviation from such assumption. Something simple like simulated normally distributed trait such as height would suffice.

-We appreciate this important suggestion. We have performed simulation studies to evaluate the impacts of (1) errors in estimating model parameter, (2) assortative mating, (3) gene-by-environment and gene-by-gene interaction effects, and (4) rare variants leading to non-normal distribution of the under-modelled genetic component. These simulations are described in detail in Supplementary Notes.

In summary, we found that our method can tolerate mild errors in parameter estimation as well as moderate violations of model assumptions above, which strongly supports its utility.

Please include potential future development/improvement in the discussion section

-We have included discussions of future development in Discussion as described above.

Finally, this reviewer would like to congratulate the author for the wonderful work presented here.

Best Regards,

Roberto Bonelli, PhD
Research Officer
Walter & Eliza Hall Institute of Medical Research

-Again, we thank Dr. Bonelli for these helpful comments and suggestions.

REVIEWERS' COMMENTS:

Reviewer #1 (Remarks to the Author):

The authors have answered all my comments (in fact, I am very impressed by the level of detail and care in their responses, thank you)

Reviewer #2 (Remarks to the Author):

The authors have addressed most of my concerns, and the manuscript has been improved. I have no further comments.

Reviewer #3 (Remarks to the Author):

The authors have addressed all the concerns presented.

This reviewer does not have any further comments to present.

Best regards,

Roberto Bonelli, PhD
WEHI